# Quality and Integrated Service Delivery: A Cross-Sectional Study of the Effects of Malaria and Antenatal Service Quality on Malaria Intervention Use in Sub-Saharan Africa

**DOI:** 10.3390/tropicalmed7110363

**Published:** 2022-11-09

**Authors:** Elizabeth H. Lee, James D. Mancuso, Tracey Koehlmoos, V. Ann Stewart, Jason W. Bennett, Cara Olsen

**Affiliations:** 1Department of Pediatrics, The Uniformed Services University of the Health Sciences, Bethesda, MD 20814, USA; 2Department of Preventive Medicine and Biostatistics, The Uniformed Services University of the Health Sciences, Bethesda, MD 20814, USA; 3Center for Health Services Research, The Uniformed Services University of the Health Sciences, Bethesda, MD 20814, USA; 4Multidrug-Resistant Organism Repository & Surveillance Network, Walter Reed Army Institute of Research, Silver Spring, MD 20910, USA

**Keywords:** malaria, pregnancy, antenatal care, service integration, quality, Sub-Saharan Africa

## Abstract

Using regionally linked facility and household surveys, we measured the quality of integrated antenatal care and malaria in pregnancy services in Kenya, Namibia, Senegal, and Tanzania. We examined country heterogeneities for the association of integrated antenatal and malaria service quality scores with insecticide-treated bed net (ITN) use in pregnant women and children under-five and intermittent preventive treatment in pregnancy (IPTp-2) uptake. Malaria in pregnancy service quality was low overall. Our findings suggest modest, positive associations between malaria in pregnancy quality and ITN use and IPTp-2 uptake across pooled models and for most studied countries, with evidence of heterogeneity in the strength of associations and relevant confounding factors. Antenatal care quality generally was not associated with the study outcomes, although a positive interaction with malaria in pregnancy quality was present for pooled ITN use models. The improved quality of malaria services delivered during formal antenatal care can help address low coverage and usage rates of preventive malaria interventions in pregnancy and childhood. Study findings may be used to target quality improvement efforts at the sub-national level. Study methods may be adapted to identify low-performing facilities for intervention and adaption to other areas of care, such as HIV/AIDS, child immunizations, and postnatal care.

## 1. Introduction

Malaria morbidity and mortality due to infection with *Plasmodium falciparum* are greatest in pregnant women, neonates, and children under five in Sub-Saharan Africa and continue to be a pressing concern for these at-risk populations [1,2]. Children often present with clinical illness due to little or no acquired immunity, which develops over time with repeated infection [3]. Infection in pregnancy may lead to novel gene expression by *P. falciparum*-infected red blood cells, placental sequestration, and inflammation and disrupted nutrient and blood flow to the fetus, with the possibility of spontaneous abortion, intrauterine growth retardation, and low birth weight [4,5,6,7,8,9,10]. Annually, an estimated 33 million African pregnancies are at risk, with roughly a third affected by gestational malaria [11,12]. Regional estimates of peripheral and placental infection range from 29.5 to 35.1 percent and 26.5 to 38 percent, respectively [13]. *P. falciparum* infection during pregnancy causes an estimated 11 percent of neonatal mortality in Sub-Saharan Africa [14]. Additionally, approximately 24 million Sub-Saharan African children are infected annually with *P. falciparum*, with an estimated 481,000 malaria deaths in children under five [11,15].

Integrated service delivery has been called for as a key component for health system strengthening activities [16]. While the available evidence is mixed [17,18,19], well-integrated services are expected to lead to the improved quality of malaria service delivery and, in turn, improved health outcomes [17,20,21,22]. The antenatal care (ANC) platform has been used for over two decades to deliver malaria in pregnancy (MiP) interventions and counseling [23], with the potential to positively influence mothers’ health behaviors and those of their children and avert thousands of deaths annually [24]. Yet, malaria service coverage during ANC has proven problematic, with lower proportions than for all other antenatal services [25].

Relatively little is known about how the facility delivery of integrated ANC MiP services translates broadly into health systems performance. Facility-level structural barriers, such as drug stock-outs, user fees, and the availability of malaria guidelines, can impact the services provided [26,27]. Small-scale studies have also demonstrated the impact of provider effort and fidelity to accepted practice standards on the patient uptake of malaria interventions in pregnancy [27,28,29,30,31]. The content of services and the way in which services are delivered are critical, yet no consensus exists on how to best measure MiP service quality—a challenge germane to quality improvement endeavors in low and middle-income countries [32].

To address preventable morbidity and mortality in mothers, infants, and children through shortfalls in the coverage and use of malaria interventions, we need to be able to systematically assess the quality of integrated services. Our study objectives were (i) to develop a theory-derived score to measure the quality of malaria services delivered during antenatal care; (ii) to determine whether the quality of integrated antenatal and malaria services predict the malaria intervention use of insecticide-treated bed nets (ITNs) and two doses of intermittent preventive treatment in pregnancy (IPTp-2), during pregnancy, and in children under five; and (iii) to document between-country variations in factors associated with ITN use and IPTp-2 uptake.

## 2. Materials and Methods

### 2.1. Data Sources

We analyzed publicly available, cross-sectional, geo-located Demographic and Health Survey (DHS) and Service Provision Assessment (SPA) data from USAID’s Demographic and Health Surveys Program [33], malaria endemicity data from the Malaria Atlas Project (MAP) [34], and regional population estimates available from the national statistics departments for Kenya, Namibia Senegal, and Tanzania. DHS household surveys were sampled within regions for household clusters and individuals. SPA data included an inventory, antenatal care observations, and health worker interviews for each health facility. MAP data provided annual, continuous malaria endemicity estimates, or *P. falciparum* prevalence rates (PfPR) standardized to 2–10 years [35]. We assigned PfPR values to DHS survey clusters using latitude and longitude coordinates in ArcGIS 10.3.

We included countries with five or more regions, 2006 data or later collected after national IPTp-2 policy adoption, and for which the collection of SPA and DHS data were proximal. Facilities offering malaria and antenatal services with complete case data were included. Women aged 15–49 years with a prior live birth in the last 24 months, children aged 0–59 months, and currently pregnant women who had complete data were included in the IPTp-2, child ITN, and pregnancy ITN analyses, respectively. ITN analyses were further restricted to households with at least one ITN. Our outcomes were (i) the receipt of two or more doses of IPTp with sulfadoxine-pyrimethamine (IPTp-SP2) during mother’s last live birth in the preceding 24 months; (ii) the prior night’s ITN use by pregnant women; and (iii) the prior night’s ITN use by children under five. Although current IPTp guidelines call for three or more SP doses (IPTp-SP3+), we used the guideline for at least two doses in effect during data collection. A list of variables according to the data source is available in Appendix A.

### 2.2. Ethical Considerations

The study protocol was reviewed, and it was deemed the research of non-human subjects by the institutional review board of the Uniformed Services University of the Health Sciences, Bethesda, MD, USA. All data were publicly available and de-identified.

### 2.3. Quality Score Development

We generated two continuous quality scores for ANC and MiP from a scale of 0 to 100 (Appendix A). We first mapped quality indicators from the literature to a theory-derived, multi-dimensional, multi-domain tool. The tool combines the World Health Organization (WHO) quality framework of six dimensions and Donabedian’s structure-process-outcomes domains of quality to aid in the systematic selection of a comprehensive, parsimonious set of indicators [36,37]. We mapped indicators of malaria service quality for services routinely conducted during antenatal care to our quality framework tool, emphasizing MiP process indicators (Table 1 and Appendix A). We calculated unweighted averages for five quality dimensions from the tool [36] and an average of dimensions to arrive at overall ANC and MiP scores. We could not construct the sixth dimension, equity, from the available SPA indicators [38]. Finally, we regionally aggregated the weighted ANC and MiP quality scores.

### 2.4. Statistics

We calculated unweighted counts with frequencies and medians with interquartile ranges. We examined the crude associations and potential confounders of ANC and MiP quality and the respective study outcomes. As data were multilevel (individual, survey cluster, region) and nationally representative, we built empty pooled and country models, which included random effects for the region and cluster levels only, to determine at what level, if any, random effects should be included. We then built pooled, adjusted, mixed effects multilevel modified Poisson models for each non-rare outcome with countries weighted equally, as well as for stratified country analyses [58,59]. Models included random effects at region and/or cluster levels, and all other variables were treated as fixed effects based on empty model results [60]. To account for the stratified survey design, we adjusted for residence location. We explored interactions between the location and mean-centered quality scores and between the two mean-centered quality scores in all models [61]. For children’s ITN use, we also assessed quality variations by the child’s sex and age. We tested model assumptions, ran goodness-of-fit tests, and performed a priori stratified analyses by country for each outcome (Appendix A) [62,63,64,65]. All analyses were conducted in Stata 14.0.

## 3. Results

Pooled analytic sample sizes for the IPTp-2, pregnancy ITN, and child’s ITN use analyses were 15,715 women, 2378 pregnant women, and 27,217 children, respectively. Unweighted malaria prevalence estimates were 6.16–7.3% on average, with substantial cross-country variation (Table 2). Country ITN and IPTp-2 use similarly varied. Median MiP quality was generally low, with Namibia being the lowest performing. Median ANC quality was moderately high for the pooled and country data, except for Tanzania. Namibia had a greater facility density, higher HIV prevalence in reproductive-age women, and more highly educated mothers. The country and pooled characteristics were otherwise similar (Appendix A).

### 3.1. IPTp-2 Uptake in Pregnancy

Pooled crude results for IPTp-2 uptake indicated a weak, positive association with MiP quality (Figure 1, Appendix A) which held for the adjusted model (Table 3). Although there was a negative crude association of ANC quality with IPTp-2 uptake, after adjustment, ANC quality had no significant effect. Stratified analyses for Kenya, Namibia, and Tanzania were generally consistent with the pooled results for MiP quality. ANC quality was negatively associated with ITPp-2 for Kenyan and Tanzanian regions with an average MiP quality; there was no association in Senegal or Namibia. In Kenya and Tanzania, as the ANC quality improved, the effect of MiP quality on IPTp-2 uptake weakened. Urban locations were consistently associated with IPTp-2 uptake. ANC visit count positively predicted IPTp-2 uptake in all models but Namibia.

### 3.2. ITN Use in Pregnancy

The results of the pooled crude analysis for pregnancy ITN use indicated a modest, positive association with MiP and an inverse association with ANC (Figure 2, Appendix A). After adjustment, a modest, positive effect of the MiP quality remained for regions with an average ANC quality (Table 4). As the regional ANC quality improved, pregnant women were more likely to use ITNs as the MiP quality improved. However, in regions with a below-average MiP quality, ITN use was inversely related to the ANC quality. Country results also generally suggested a modest effect of the MiP quality on ITN use in pregnancy for Kenya and Namibia and for regions with an average ANC quality in Senegal. In Namibia, the ANC quality was negatively associated with pregnancy ITN use. A positive, nonlinear interaction between the ANC and MiP quality in Senegal indicated an inverse relationship between the MiP quality and ITN use for regions with a below-average ANC quality. In Senegalese regions with above-average ANC quality, the likelihood of individual ITN use increased with improved MiP quality. As the malaria burden increased, pregnancy ITN use tended to increase in the pooled Kenya and Tanzania models. Additionally, the effect of urban location varied directionally by country.

### 3.3. ITN Use in Children Under-Five

The pooled crude and adjusted results for children’s ITN use suggested a modest association between regional MiP quality and the child’s ITN use (Figure 3, Table 5 and Appendix A). We identified a curvilinear, significant interaction between ANC and MiP quality. There was little discernable relationship in regions with a below-average ANC quality, but there was a positive relationship between the MiP quality and ITN use in regions with at least average ANC quality. In regions with both a high MiP and ANC quality, there was an inverse relationship with ITN use.

MiP quality and children’s ITN use were modestly associated in all countries and were significant for Senegal and Kenya. In Kenya, the strength of this relationship increased as malaria endemicity increased. ANC was not associated with a child’s ITN use. In Kenyan and Senegalese regions with an average ANC quality, urban children were more likely to use a net. A weak, inverse relationship was apparent in rural areas as ANC quality improved. HIV prevalence in women of reproductive age was the sole significant predictor for a child’s ITN use in Namibia. Children’s ITN use increased with the malaria burden in the pooled Kenya and Tanzania models.

## 4. Discussion

We found low MiP service quality for all countries using our indicator set. We saw consistent, albeit modest, adjusted effects of MiP service quality across the pooled and country-specific models for all the study outcomes (Figure 4). Country ANC quality scores were somewhat higher overall than for MiP quality, although cautious comparisons of the ANC and MiP quality scores are warranted, given non-identical services. After controlling for potential confounders and regional and cluster effects in the pooled models, there was no relationship between the average ANC quality and ITN outcomes.

The relationship between ANC and MiP quality varied for the pooled and stratified models and by the outcome. A significant, positive interaction between ANC and MiP quality was present in our pooled ITN use models and for stratified Senegal ITN use in pregnancy. A negative interaction between ANC and MiP quality was present for the Kenya and Tanzania IPTp-2 models but not for the pooled IPTp-2 model.

Country variations in the confounders and effect modifiers of ANC or MiP quality and the study outcomes were also present. For example, in Kenya, the facility density was an important confounder of MiP quality and children’s usage of ITNs but was not important for the pooled or other country models. In Kenya, urban households of an average ANC quality were more likely to report the use of an ITN in pregnancy or childhood, but in Senegal, this was only true for children.

Our findings support existing evidence of the need for high-quality integrated ANC and MiP services to improve health outcomes [14,53,66]. Although most countries have rolled out the ANC package, poor national coordinating and planning mechanisms for integration and non-functional quality assurance systems may remain [67]. In Kenya, ITNs are more readily delivered via ANC than IPTp, highlighting the need for improved IPTp services [53]. Improved malaria knowledge has had a positive influence on both IPTp and ITN use in pregnancy, including via group ANC education sessions [68,69,70]. In Tanzania, the ability to integrate malaria services was significantly improved over tuberculosis and HIV services, and having staff trained in infectious diseases on-site significantly predicted the likelihood of receiving integrated services during ANC [71].

Consistent with previous works, we found that ANC visits positively predicted IPTp-2 but not ITN use for pooled and most country data [50]. The urban location results aligned with previous findings for IPTp uptake as well as ITN use in childhood; however, they were inconsistent for ITN use in pregnancy across the stratified and pooled models [50]. Although we could not assess for a reciprocal effect on ANC quality given the longstanding integration through the ANC package, the presence of MiP programming might be expected to have a positive effect on ANC quality, as has been similarly demonstrated for HIV programming [72].

### 4.1. Strengths and Limitations

Our study demonstrates an approach for linking routinely-collected facility and household survey data when individuals cannot be directly associated with the facilities where care was sought [73]. Current repositories of routine and nationally representative data offer an alternative method for quality assessment rather than primary data collection [74] when answers to data-suitable research questions are needed. Our approach demonstrates the possibility of combining these data for low-cost, high-yield results in relation to the contemporary issue of service integration and in response to the need for baseline health systems strengthening data and new health systems research methods [75]. Furthermore, this method allows for within- and cross-country health systems performance comparisons.

We also highlight how the selection of quality metrics can be systematic and theory -driven. We operationalized a novel tool that builds on prior work [32,38] to ensure the representation of multiple, well-accepted quality dimensions [36] while simultaneously selecting indicators across the structure, process, and outcome continuum [37]. To our knowledge, this is the first study of antenatal care and malaria in pregnancy that combines these frameworks for joint operationalization.

Our work had several limitations. Residual confounding within these secondary datasets may remain for several reasons. These include an inability to measure certain individual-level predictors, e.g., the number of ANC visits during a current pregnancy, and an inability to include certain higher-order confounders. For example, a range of external governance, financial, policy, human resource, supply chain, and information systems challenges could affect the coverage and uptake of interventions [23,50,76]. Indeed, the integration of often-siloed governance structures, funding, and services holistically ensures maternal and child health outcomes [77]. Additionally, although DHS data were not commonly missing in this study, we cannot rule out the possibility of bias introduced through the missingness of SPA data used to construct quality indices in the event that this was due to systematic non-response. Although regionally aggregated for our study, residual confounding may remain [38].

While an important alternative to more costly and complex survey designs, aggregating facility-weighted scores to link data regionally results in a loss of variation. The feasibility of the approach depends on sufficient linkage level variability, and this may require certain country exclusions, e.g., Malawi, with only three regions. Our approach may be best reserved for cross-country comparisons to identify performance gaps.

Finally, a cautious interpretation of the findings is warranted. Cross-sectional data prevents assessment for causality. Additionally, small sample sizes may have played a role in nonsignificant findings for the stratified country models. For example, we cannot be certain whether a smaller sample size for the Namibia analysis of children’s ITN use may have limited our ability to detect an effect of MiP quality or other important effects. Still, we found important, if modest, associations between the quality and our outcomes for the country and pooled models.

### 4.2. Public Health Impact

Our findings suggest that improved delivery and education on the use of interventions continue to be integral components of malaria prevention in Sub-Saharan Africa. Generally low MiP quality in all countries indicated that broad quality of care improvements could be necessary. Strengthening existing facility-based delivery mechanisms is a means to address the gaps in the national coverage and usage targets, particularly the persistently low MiP targets.

Strengthening facility delivery mechanisms will require evaluation tools and consensus on a standardized, comprehensive, and readily measured set of indicators for MiP quality. Although the malaria components of ANC are well-established [23], we found few examples in the literature where this translated into the consistent use of standardized malaria quality indicators. Our quality tool can help ensure the systematic, comprehensive selection of relevant indicators measuring the full quality spectrum and can identify indicator gaps, e.g., for outcomes and equity measures.

We found several negative interactions between ANC and MiP service quality, e.g., for IPTp-2 uptake in Kenya and Tanzania. This could reflect improved ANC and MiP quality in cities with low endemicity, where improved malaria knowledge in well-educated populations might attenuate the effect of service quality. Indeed, our study found that higher education was positively associated with IPTp-2 uptake in Tanzania, which is consistent with local studies that demonstrate a link between educational attainment and malaria knowledge and practice [78,79]. Alternatively, provider hesitancy to prescribe prophylaxis for complex cases, such as co-infection with HIV, could play a role [80]. Nuances in the results for pooled ITN use suggest an inverse relationship for children’s ITN use in regions of both a high MiP and ANC quality and its decreased use in pregnancy as the ANC quality improved in regions with a below average MiP quality. The former may reflect affluent, urban populations with improved care access and a perceived lower threat of malaria as a rural disease of poverty, especially where prior interventions may have targeted rural areas [81,82,83]. The latter may reflect areas where there was little or perceived little risk of malaria but where the ANC quality was higher, and thus, there was little actual or perceived need to consistently deliver MiP services as a part of ANC, e.g., parts of Namibia. Further studies of the fidelity of preventive healthcare delivery components in urban areas would help tailor interventions, particularly in light of recent evidence suggesting outbreaks and/or a resurgence of malaria due to urban vectors, such as *Anopheles stephensi* [84].

The generally modest effect of MiP service quality, paired with mixed findings for the interactions between ANC and MiP, suggests a need for geographically targeted and improved integration of these services to strengthen impact. Our findings are a starting point that requires further consideration and evidence generation. An agreement on best practices for assessing integrated service delivery performance is needed. Although the integration of primary healthcare services as a tool to strengthen health systems is expected to lead to improved service delivery and health outcomes [74], there is a dearth of evidence to support this [85].

Our methods of utilizing secondary data can potentially be extended to other integrated service areas for baseline and trend analyses to address this gap. Furthermore, the adaptation of our methodology to pre- and post- analyses can be used to evaluate the progress over time toward IPTp-SP3+ policy implementation in facilities and identify low-performing facilities and country geographic areas for targeted improvement. This is particularly relevant in light of service delivery interruptions during scenarios such as the recent COVID-2019 pandemic. The use of our methods in real-time settings would require primary data collection and further testing accounting for stated limitations to demonstrate the accuracy, operational utility, feasibility, and reliability of applications by national programs.

## 5. Conclusions

The ANC package is a long-standing example of how integrated service delivery requires careful thought and consistent re-evaluation, as evidenced by 2016 updates recommending increased visits. Our findings support the continued need for high-quality, improved integrated antenatal and malaria services as a delivery channel for malaria in pregnancy interventions. Additionally, our quality assessment tool may be adapted and operationalized in a range of service delivery environments for systematic quality improvements.

## Figures and Tables

**Figure 1 tropicalmed-07-00363-f001:**
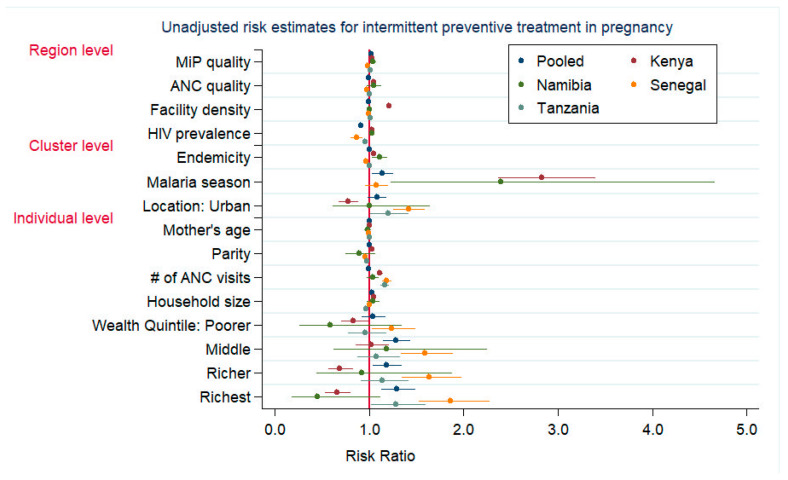
Unadjusted risk estimates for intermittent preventive treatment in pregnancy. Note: Unadjusted risk estimates of mother’s educational level not shown.

**Figure 2 tropicalmed-07-00363-f002:**
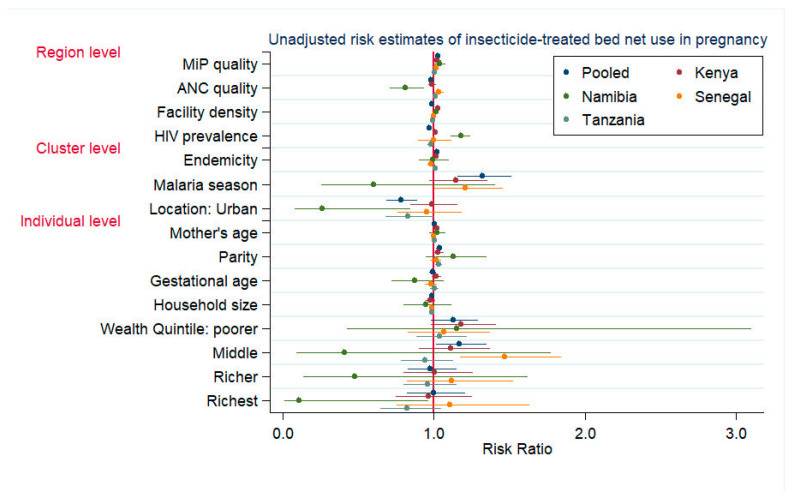
Unadjusted risk estimates for insecticide-treated bed net use in pregnancy. Note: Unadjusted risk estimates of mother’s educational level not shown.

**Figure 3 tropicalmed-07-00363-f003:**
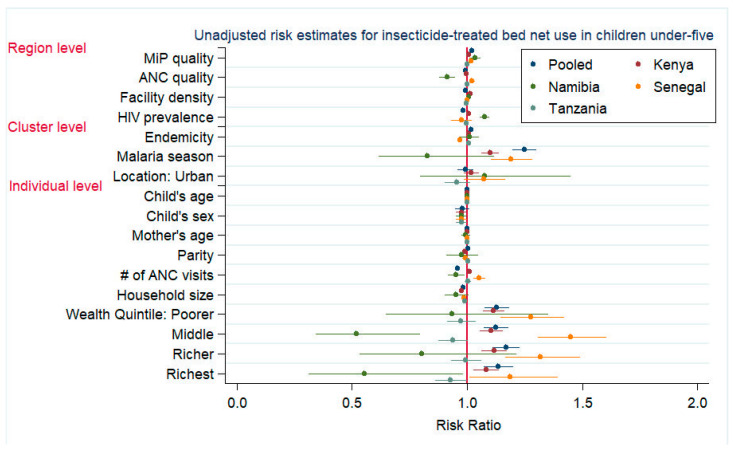
Unadjusted risk estimates for insecticide-treated bed net use in children under five. Note: Unadjusted risk estimates of mother’s educational level not shown.

**Figure 4 tropicalmed-07-00363-f004:**
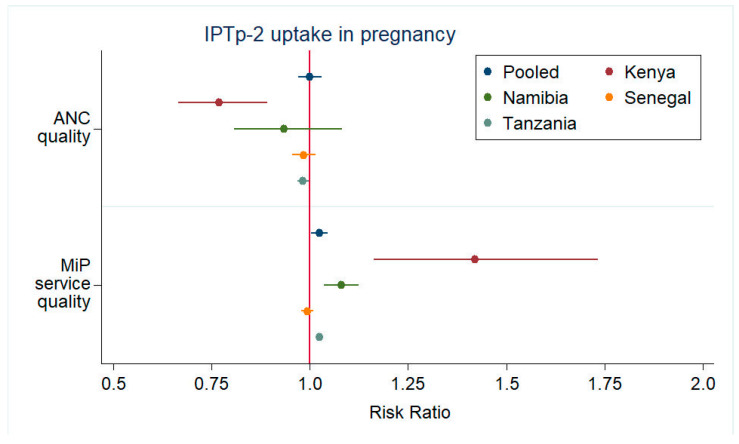
Forest plots of adjusted associations of malaria in pregnancy and antenatal care quality with each of three study outcomes. Abbreviations: IPTp-2—intermittent preventive treatment in pregnancy—2 doses; ITN—insecticide-treated bed net; ANC—antenatal care; MiP—malaria in pregnancy.

**Table 1 tropicalmed-07-00363-t001:** Mapping of malaria in pregnancy quality indicators to combined quality framework tool.

		World Health Organization Framework					
		Effectiveness	Efficiency	Accessibility	Acceptability/ Patient-Centeredness	Safety	Equity
**Donabedian Framework**	**Structure**	- Guidelines on IPTp observed [26,39,40].	- Frequency of routine meetings for reviewing managerial or administrative matters [41].	- Country first-line treatment available [39,42,43,44,45]. - Valid SP/Fansidar observed available [26,30,39]. - mRDT or microscopy observed with all components functional: valid RDT OR microscopy: light microscope, glass slides, covers, stain [39,42,43,46,47]. - ITN observed in stock [39,48].	- Medication fees for medications given during ANC OR general fees for medications other than ARV therapy OR fees for IPTp-SP [26,49].	- On-duty provider ever received any pre-service or in-service training on IPTp [26,39,50]. - On-duty provider ever received any in-service training or training updates on pregnancy complications and management [51].	
	**Process**	- IPTp reported as routinely offered during ANC [26,45]. - Provider prescribed or gave anti-malarial prophylaxis [52]. - Importance of a further dose of IPT explained [53]. - Screening for anemia occurred if: tested hemoglobin levels AND asked client about tiredness or breathlessness AND provider asked or client mentioned fever, headache/blurred vision [3,39,40].	- Explained how to take the anti-malarial medications [54]. - Observed that the [1st] dose of IPTp is given in the facility [26,40,45,52].	- Provided ITN free of charge or voucher to client as part of consultation or instructed client to obtain ITN elsewhere in facility [45,52,53].	- Explicitly explained importance of using ITN [3,53,55]. - Explained purpose of preventive treatment with malaria medications [3,40,47,49,50].	- Explained possible side effects of malaria pills [56,57].	
	**Outcome**						

Note: Greyed out indicators were not included in the final quality score due to high correlation with other indicators. Abbreviations: IPTp—intermittent preventive treatment in pregnancy; SP—Sulfadoxine-Pyrimethamine; mRDT—malaria rapid diagnostic test; ITN—insecticide treated bed net; ANC—antenatal care; ARV—antiretroviral.

**Table 2 tropicalmed-07-00363-t002:** Selected unweighted characteristics of country and pooled data by study outcome.

Country	SPA Year	DHS, MAP and Pop. Year	N	Outcome	Outcome Prevalence ^1^	MiP Quality ^2^	ANC Quality ^2^	Malaria Endemicity ^2^	Facility Density ^2^
Kenya	2010	2014	7861	IPTp-2	1409 (17.92)	54.47 (53.32, 54.51)	76.32 (74.56, 79.18)	9.02 (4.50, 18.91)	12.01 (6.41, 14.64)
			662	ITN in pregnancy	470 (71.00)	54.47 (53.32, 54.51)	76.32 (74.56, 79.18)	11.03 (5.90, 21.23)	12.01 (6.41, 15.54)
			13,870	ITN in children under-5	10,073 (72.62)	54.47 (53.32, 54.51)	76.32 (74.56, 79.18)	11.09 (5.99, 21.38)	12.01 (6.41, 15.54)
Namibia	2009	2013	1639	IPTp-2	69 (4.21)	30.18 (16.95, 35.64)	73.05 (69.59, 74.35)	5.17 (0.00,7.64)	131.14 (86.12, 184.40)
			207	ITN in pregnancy	23 (11.11)	35.64 (30.18, 38.87)	74.24 (70.03, 77.79)	6.55 (2.37, 7.86)	96.38 (86.12, 184.40)
			1570	ITN in children under-5	250 (15.92)	33.45 (27.64, 37.13)	73.86 (69.59, 76.10)	6.24 (1.83, 7.79)	130.61 (92.90, 184.40)
Senegal	2014	2013	2682	IPTp-2	1048 (39.08)	44.51 (42.81, 47.05)	73.56 (71.23, 76.64)	2.54 (2.05, 3.69)	24.60 (20.53, 32.14)
			729	ITN in pregnancy	400 (54.87)	44.51 (42.81, 47.05)	75.09 (71.23, 76.64)	2.46 (1.96, 3.39)	24.60 (20.53, 33.34)
			5602	ITN in children under-5	3172 (56.62)	46.38 (42.81, 50.00)	73.56 (71.23, 76.64)	2.46 (1.99, 3.39)	24.60 (20.53, 33.34)
Tanzania	2006	2010	2993	IPTp-2	990 (33.08)	42.26 (38.97, 46.33)	57.83 (53.70, 63.26)	7.13 (4.34, 13.68)	9.30 (6.48, 13.91)
			780	ITN in pregnancy	515 (66.03)	42.70 (38.97, 46.33)	56.64 (53.70, 63.66)	8.12 (4.39, 14.95)	9.16 (6.48, 13.91)
			6175	ITN in children under-5	4300 (69.64)	42.26 (38.97, 46.33)	57.83 (53.70, 63.66)	7.40 (4.35, 14.29)	9.30 (6.48, 13.91)
Pooled	--	--	15,175	IPTp-2	3516 (23.17)	52.94 (41.94, 54.51)	74.56 (70.03, 76.90)	6.69 (2.84, 13.03)	14.62 (9.02, 20.96)
			2378	ITN in pregnancy	1408 (59.21)	46.33 (41.38, 53.32)	73.37 (63.66, 76.64)	6.16 (2.65, 12.55)	15.54 (9.30, 28.05)
			27,217	ITN in children under-5	17,795 (65.38)	52.94 (42.81, 54.51)	74.56 (69.82, 76.90)	7.30 (3.07, 15.45)	13.91 (9.13, 20.53)

Abbreviations: SPA—service provision assessments; DHS—demographic and health surveys; MAP—Malaria Atlas Project; Pop.—population; N—count; MiP—malaria in pregnancy; ANC—antenatal care; IPTp-2—intermittent preventive treatment in pregnancy—2 doses; ITN—insecticide-treated bed net. Note: ^1^ Outcome prevalence given as count with percentage in parenthesis. ^2^ Median value with and inter-quartile range in parenthesis.

**Table 3 tropicalmed-07-00363-t003:** Adjusted multilevel mixed effects modified Poisson results for intermittent preventive treatment in pregnancy uptake.

		Kenya (n = 7861)		Namibia (n = 1639)		Senegal (n = 2682)		Tanzania (n = 2993)		Pooled (n = 15,175)	
		RR	95% CI	RR	95% CI	RR	95% CI	RR	95% CI	RR	95% CI
Adjusted Model ^1^	Measures of Association										
Individual Level	Number of ANC visits	**1.136**	**(1.100, 1.173)**	--	--	**1.169**	**(1.085, 1.259)**	**1.147**	**(1.103, 1.192)**	**1.127**	**(1.092, 1.163)**
	Mother’s education (none)	--	--	--	--	Ref	Ref	Ref	Ref	--	--
	Primary	--	--	--	--	1.131	(0.856, 1.493)	1.164	(0.999, 1.355)	--	--
	Secondary or higher	--	--	--	--	**0.832**	**(0.712, 0.972)**	**1.553**	**(1.203, 2.005)**	--	--
Cluster Level	Residence location (urban)	1.014	(0.844, 1.219)	**1.482**	**(1.015, 2.164)**	**1.187**	**(1.056, 1.335)**	1.093	(0.889, 1.345)	**1.133**	**(1.031, 1.244)**
Region Level	ANC quality ^3^	**0.771**	**(0.665, 0.892)**	0.935	(0.807, 1.082)	0.984	(0.955, 1.015)	**0.984**	**(0.970, 0.997)**	1.000	(0.971, 1.031)
	MiP quality ^3^	**1.420**	**(1.163, 1.733)**	**1.080**	**(1.037, 1.126)**	0.994	(0.978, 1.010)	**1.026**	**(1.016, 1.035)**	**1.024**	**(1.004, 1.045)**
	Facility density ^4^	**1.508**	**(1.208, 1.883)**	**0.993**	**(0.986, 0.999)**	--	--	**1.007**	**(1.003, 1.011)**	--	--
	HIV prevalence ^5^	1.028	(0.928, 1.138)	--	--	--	--	**0.969**	**(0.952, 0.987)**	--	--
	ANC quality × MiP quality ^6^	**0.939**	**(0.893, 0.987)**	--	--	--	--	**0.998**	**(0.997, 0.999)**	--	--
Country Level	Country (Kenya)	**--**	**--**	**--**	**--**	**--**	**--**	**--**	**--**	Ref	Ref
	Namibia	**--**	**--**	**--**	**--**	**--**	**--**	**--**	**--**	0.437	(0.148, 1.291)
	Senegal	**--**	**--**	**--**	**--**	**--**	**--**	**--**	**--**	**4.251**	**(2.000, 9.036)**
	Tanzania	**--**	**--**	**--**	**--**	**--**	**--**	**--**	**--**	**3.611**	**(1.651, 7.897)**
	Measures of Variation										
	Region level	**0.281**	**(0.130, 0.607)**	**0.255**	**(0.105, 0.620)**	**0.011**	**(0.003, 0.045)**	**0.011**	**(0.002, 0.071)**	**0.235**	**(0.130, 0.427)**
	Cluster level	**0.109**	**(0.024, 0.494)**	0.064	(0.000, 56,779.11)	--	--	**0.066**	**(0.024, 0.180)**	**0.070**	**(0.033, 0.149)**
	**Model AIC**	58,891.522		607.330		3821.163		3852.737		15,921.980	
Empty Model ^2^	Measures of Variation										
	Region level	1.244	(0.642, 2.410)	**0.458**	**(0.233, 0.899)**	**0.043**	**(0.019, 0.099)**	**0.052**	**(0.024, 0.115)**	1.019	(0.668, 1.557)
	Cluster level	**0.117**	**(0.020, 0.681)**	0.121	(0.000, 74.279)	0.003	(0.00, 1059.45)	**0.092**	**(0.042, 0.199)**	**0.094**	**(0.050, 0.178)**
	Model AIC	5958.706		607.0791		3887.345		3899.29		16,142.320	

Abbreviations: RR—risk ratio; CI—confidence interval; Ref—reference level; ANC—antenatal care; MiP—malaria in pregnancy; AIC—Akaike’s information criterion. Note: Bolded numbers indicate statistical significance. ^1^ Adjusted model: random coefficient of clusters and/or regions with remaining significant variables after adjustment. ^2^ Empty model: solely random coefficient of clusters and/or regions. ^3^ Mean-centered for each country and overall, for pooled model. ^4^ Per 1,000,000 population. ^5^ In women ages 15–49. ^6^ Calculated using mean-centered quality score(s).

**Table 4 tropicalmed-07-00363-t004:** Adjusted multilevel mixed effects modified Poisson results for insecticide-treated net use in pregnancy.

		Kenya (n = 662)		Namibia (n = 586)		Senegal (n = 729)		Tanzania (n = 780)		Pooled (n = 2378)	
		RR	95% CI	RR	95% CI	RR	95% CI	RR	95% CI	RR	95% CI
Adjusted Model ^1^	Measures of Association										
Individual Level	Mother’s age (in years)	**1.019**	**(1.014, 1.025)**	--	--	--	--	--	--	--	--
	Parity	--	--	--	--	--	--	**1.035**	**(1.011, 1.060)**	--	--
	Household size	**--**	**--**	--	--	--	--	**0.978**	**(0.963, 0.992)**	--	--
Cluster Level	Malaria endemicity	**1.012**	**(1.007, 1.018)**	--	--	--	--	**1.009**	**(1.003, 1.016)**	**1.009**	**(1.004, 1.014)**
	Residence location (urban)	**1.174**	**(1.062, 1.298)**	**0.393**	**(0.178, 0.869)**	1.068	(0.844, 1.352)	**0.825**	**(0.686, 0.993)**	0.985	(0.873, 1.112)
Region Level	ANC quality ^3^	0.987	(0.964, 1.011)	**0.809**	**(0.755, 0.866)**	1.022	(0.974, 1.073)	1.013	(0.996, 1.030)	0.999	(0.978, 1.021)
	MiP quality ^3^	**1.022**	**(1.018, 1.026)**	**1.080**	**(1.007, 1.157)**	1.008	(0.988, 1.028)	0.997	(0.992, 1.003)	**1.014**	**(1.003, 1.025)**
	Facility density ^4^	**1.034**	**(1.013, 1.054)**	--	--	--	--	--	--	--	--
	ANC quality × residence location ^5^	**--**	**--**	--	--	**--**	**--**	--	--	--	--
	ANC quality × MiP quality ^5^	**--**	**--**	--	--	**1.006**	**(1.001, 1.012)**	--	--	**1.001**	**(1.001, 1.002)**
Country Level	Country (Kenya)	--	--	--	--	**--**	**--**	--	--	Ref	Ref
	Namibia	--	--	--	--	**--**	**--**	--	--	**0.230**	**(0.110, 0.482)**
	Senegal	--	--	--	--	**--**	**--**	--	--	0.957	(0.771, 1.188)
	Tanzania	--	--	--	--	**--**	**--**	--	--	1.210	(0.788, 1.858)
	Measures of Variation										
	Region level	**0.000**	**(0.000, 0.000)**	**0.000**	**(0.000, 0.000)**	0.938	(0.006, 0.219)	**0.000**	**(0.000, 0.000)**	0.032	(0.010, 0.103)
	Cluster level	--	--	--	--	0.060	(0.003, 1.130)	--	--	0.025	(0.001, 0.736)
	Model AIC	1191.628		128.191		1096.993		1503.685		4840.075	
Empty Model ^2^	Measures of Variation										
	Region level	**0.027**	**(0.007, 0.109)**	**0.761**	**(0.169, 3.419)**	**0.048**	**(0.014, 0.163)**	0.000	(0.000, 1.3 × 10^155^)	0.048	(0.171, 1.324)
	Cluster level	--	--	--	--	0.063	(0.004, 1.016)	--	--	**0.029**	**(0.119, 0.692)**
	Model AIC	1204.449		140.769		1090.682		1506.818		2774.930	

Abbreviations: RR—risk ratio; CI—confidence interval; Ref—reference level; ANC—antenatal care; MiP—malaria in pregnancy; AIC—Akaike’s information criterion. Note: Bolded numbers indicate statistical significance. ^1^ Adjusted model: random coefficient of clusters and/or regions with remaining significant variables after adjustment. ^2^ Empty model: solely random coefficient of clusters and/or regions. ^3^ Mean-centered for each country and overall, for pooled model. ^4^ Per 1,000,000 population. ^5^ Calculated using mean-centered quality score(s).

**Table 5 tropicalmed-07-00363-t005:** Adjusted multilevel mixed effects modified Poisson results for insecticide-treated net use in children under five.

		Kenya (n = 13,870)		Namibia (n = 1570)		Senegal (n = 3729)		Tanzania (n = 6175)		Pooled (n = 27,217)	
		RR	95% CI	RR	95% CI	RR	95% CI	RR	95% CI	RR	95% CI
Adjusted Model ^1^	Measures of Association										
Individual Level	Mother’s education (none)	Ref	Ref	Ref	Ref	Ref	Ref	Ref	Ref	Ref	Ref
	Primary	**--**	**--**	**--**	**--**	**--**	**--**	**--**	**--**	1.030	(0.993, 1.068)
	Secondary or higher	**--**	**--**	**--**	**--**	**--**	**--**	**--**	**--**	**1.131**	**(1.063, 1.204)**
	Number of ANC visits	**--**	**--**	**--**	**--**	**1.025**	**(1.000, 1.051)**	**--**	**--**	**--**	**--**
	Household size	**--**	**--**	**--**	**--**	**--**	**--**	**0.981**	**(0.969, 0.994)**	**0.981**	**(0.976, 0.986)**
Cluster Level	Malaria endemicity	**1.003**	**(1.000, 1.007)**	--	--	**--**	**--**	**1.005**	**(1.002, 1.009)**	**1.005**	**(1.001, 1.009)**
	Survey timing	**1.044**	**(1.003, 1.087)**	--	--	**--**	**--**	**--**	**--**	**--**	**--**
	Residence location (urban)	**1.108**	**(1.071, 1.146)**	1.293	(0.964, 1.734)	**1.208**	**(1.096, 1.331)**	1.003	(0.963, 1.045)	**1.113**	**(1.056, 1.174)**
Region Level	ANC quality ^3^	0.996	(0.988, 1.004)	0.974	(0.893, 1.062)	1.026	(0.991, 1.062)	0.998	(0.985, 1.010)	0.988	(0.975, 1.002)
	MiP quality ^3^	**1.024**	**(1.015, 1.033)**	1.029	(0.993, 1.067)	**1.019**	**(1.001, 1.037)**	1.001	(0.995, 1.007)	**1.021**	**(1.010, 1.032)**
	Facility density ^4^	**1.020**	**(1.017, 1.023)**	--	--	**--**	**--**	--	--	--	--
	HIV prevalence ^5^	**--**	**--**	**1.081**	**(1.031, 1.133)**	**--**	**--**	--	--	--	--
	Endemicity × MiP quality ^6^	**1.001**	**(1.001, 1.002)**	--	--	--	--	--	--	--	--
	ANC quality × location ^6^	**0.988**	**(0.981, 0.994)**	--	--	**0.944**	**(0.924, 0.964)**	--	--	--	--
	ANC quality × MiP quality ^6^	--	--	--	--	--	--	--	--	**1.001**	**(1.001, 1.002)**
Country Level	Country (Kenya)	--	--	--	--	**--**	**--**	--	--	Ref	Ref
	Namibia	--	--	--	--	**--**	**--**	--	--	**0.279**	**(0.174, 0.448)**
	Senegal	--	--	--	--	**--**	**--**	--	--	1.113	(0.899, 1.379)
	Tanzania	--	--	--	--	**--**	**--**	--	--	1.051	(0.780, 1.418)
	*Measures of Variation*										
	Region level	**0.000**	**(0.000, 0.000)**	**0.106**	**(0.031, 0.362)**	**0.035**	**(0.011, 0.110)**	**0.015**	**(0.008, 0.026)**	**0.063**	**(0.034, 0.119)**
	Cluster level	--	--	0.236	(0.043, 1.303)	**0.062**	**(0.025, 0.154)**	--	--	**0.023**	**(0.008, 0.066)**
	Model AIC	25,880.360		1255.409		5719.808		11,985.670		42,520.680	
Empty Model ^2^	*Measures of Variation*										
	Region level	**0.005**	**(0.003, 0.012)**	0.524	(0.220, 1.247)	**0.052**	**(0.024, 0.114)**	**0.018**	**(0.011, 0.031)**	0.652	(0.340, 1.250)
	Cluster level	--	--	0.223	(0.042, 1.168)	**0.149**	**(0.087, 0.254)**	--	--	**0.028**	**(0.010, 0.073)**
	Model AIC	25,927.490		1263.894		8458.662		12,005.910		42,803.670	

Abbreviations: RR—risk ratio; CI—confidence interval; Ref—reference level; ANC- antenatal care; MiP—malaria in pregnancy; AIC—Akaike’s information criterion. Note: Bolded numbers indicate statistical significance. ^1^ Adjusted model: random coefficient of clusters and/or regions with remaining significant variables after adjustment. ^2^ Empty model: solely random coefficient of clusters and/or regions. ^3^ Mean-centered for each country and overall, for pooled model. ^4^ Per 1,000,000 population. ^5^ In women ages 15–49. ^6^ Calculated using mean-centered quality score(s).

## Data Availability

All datasets utilized were publicly accessible at the time of download. The demographic and health survey (DHS) and service provision assessment (SPA) data from USAID’s Demographic and Health Surveys Program are available upon request for research purposes here: https://dhsprogram.com/data/available-datasets.cfm, accessed on 1 August 2016. Malaria endemicity data from the Malaria Atlas Project (MAP) are available here: https://malariaatlas.org/, accessed on 20 August 2015. Regional population estimates were accessed from national statistics department websites for Kenya, Namibia Senegal, and Tanzania.

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
