# Peer review of "Quality and Integrated Service Delivery: A Cross-Sectional Study of the Effects of Malaria and Antenatal Service Quality on Malaria Intervention Use in Sub-Saharan Africa"

_tropicalmed, 2022, doi:10.3390/tropicalmed7110363_

Round 1

Reviewer 1 Report

Malaria in pregnancy in Sub-Saharan Africa is a well-recognized problem and has gained attention for a long time.  The issues related to the uptake of ITN use and IPTp are gaining momentum and a number of publications are available on its assessment using the WHO recommended indicators.

The model proposed by the authors appears to be novel, however, at the same time limitations are apparent in terms of its real-time application. The authors suggest that the quality assessment tool may be adapted and operationalized in a range of service delivery environments for systematic quality improvements which again does not appear to be possible by using the model/tool in its present form. The actual utility and accuracy of the model are possible only after testing it in real-time situations in limited settings including all parameters that were listed as limitations in this paper. Further work is necessary on this model to prove its on-the-ground utility and reliability as well as to make it operationally feasible and adaptable by the National Programs and health care providers. It is suggested that authors should comment on this aspect in the concluding paragraph in a lucid manner and give a lead to the readers about the requirements for the improvisation of this model to transform it into a reliable assessment tool.

Author Response

Thank you for your helpful review. We agree, that for real-time application our methods would require additional testing and adaptation. Our intention, however, is to suggest that these methods are useful for generating baseline information and to assess trends over time using secondary data sources, as opposed to operationalizing in a program-oriented context with real-time data. 

To address this concern, we added the following to our final paragraph in the discussion:

Our methods utilizing secondary data can potentially be extended to other integrated service areas for baseline and trend analyses to address this gap. Further, adaptation of our methodology to pre- and post- analyses can be used to evaluate progress over time toward IPTp-SP3+ policy implementation in facilities and identify low performing facility and within country geographic areas for targeted improvement. This is particularly relevant in light of service delivery interruptions during scenarios such as the recent Coronavirus-2019 pandemic. Use of our methods in real-time settings would require primary data collection and further testing accounting for stated limitations to demonstrate accuracy, operational utility, feasibility and reliability for applications by national programs.

Reviewer 2 Report

This is the first study to investigate the integrated service delivery of ANC and MiP, very interesting. I have some minor comments. The reference 32-34 showed the authors accessed data in 2015 and 2016, when the study was conducted.

At the current manuscript, the data was pooled from 4 countries. In fact, with secondary data from 4 countries, DHS, SPA etc, may be with some differences such as year of data, levels of data (nation-wide or DSS sites, full or partial data sets), who owned the data (any official permission required to access full data set, if so, the authors should add this information to the Ethical consideration), lists of variables, making a comparison table will be helpful for the readers.

In the limitation, the authors should address a potential problem (if any) with secondary data in particular to quality of data (e.g., missing data, incompatible variables) and how to deal with this problem. With the result and limitation presented, the authors should detail recommendation for further studied.  

Author Response

Dear Reviewer,

Thank you so much for your helpful feedback. We attempt to address your concerns here and in manuscript and its supplemental files.

  1. the data was pooled from 4 countries. In fact, with secondary data from 4 countries, DHS, SPA etc, may be with some differences such as year of data, levels of data (nation-wide or DSS sites, full or partial data sets), who owned the data (any official permission required to access full data set, if so, the authors should add this information to the Ethical consideration), lists of variables, making a comparison table will be helpful for the readers

We agree. Table 1 includes the data source and year of the data, and a description of each data set is included in the methods section. A mandatory statement on Data Access/Availability from original source is included at the end of the manuscript in alignment with journal requirements. We now include a new Table S1 in the supplemental files and reference it in the main manuscript that lists variables according to data source. Associated main text changes are as follows:

We included countries with: five or more regions, 2006 data or later collected after national IPTp-2 policy adoption, and for which collection of SPA and DHS data were proximal (Table 1). Facilities offering malaria and antenatal services with complete case data were included. Women 15-49 years with a prior live birth in the 24 months prior, children 0-59 months, and currently pregnant women who had complete data were included in the IPTp-2, child ITN, and pregnancy ITN analyses, respectively. ITN analyses were further restricted to households with at least one ITN. Our outcomes were (i) receipt of two or more doses of IPTp with sulfadoxine-pyrimethamine (IPTp-SP2) during last live birth in preceding 24 months; (ii) prior night’s ITN use by pregnant women; and (iii) prior night’s ITN use by children under-five. Although current IPTp guidelines call for three or more SP doses (IPTp-SP3+), we used the guideline for at least two doses in effect at data collection. A list of variables according to data source is available in Table S1.

2. In the limitation, the authors should address a potential problem (if any) with secondary data in particular to quality of data (e.g., missing data, incompatible variables) and how to deal with this problem. 

Thank you for this astute suggestion. Under 4.1 Strengths and limitations paragraph 3 we added:

Our work had several limitations. Residual confounding within secondary datasets may remain for several reasons. These include inability to measure certain individual-level predictors, e.g. number of ANC visits during a current pregnancy, or inability to include certain higher-order confounders. For example, a range of external governance, financial, policy, human resource, supply chain and information systems challenges could affect coverage and uptake of interventions [23,51,77]. Indeed, integration of often-siloed governance structures, funding and services holistically ensures maternal and child health outcomes [78]. Additionally, although DHS data were not commonly missing in this study, we cannot rule out the possibility of bias introduced through missingness of SPA data used to construct quality indices, in the event this was due to systematic non-response. Although regionally aggregated for our study, residual confounding may remain [39].

3. With the result and limitation presented, the authors should detail recommendation for further studied.  

 4.2 Public health impact, end of paragraph 3, we have added:

Further study of fidelity of preventive healthcare delivery components in urban areas would help tailor interventions, particularly in light of recent evidence suggesting outbreaks and/or a resurgence of malaria due to urban vectors such as Anopheles stephensi [85].

Reviewer 3 Report

Add gravity of the situation in introdution part. Discussion need support from local and regional evidence

Author Response

Thank you for your review. Please find references to where we have updated the text to be responsive to your helpful feedback.

1. Add gravity of the situation in introduction part.

We added the following:

I. Introduction, paragraph one: 

Malaria morbidity and mortality due to infection with Plasmodium falciparum is greatest in pregnant women, neonates and children under-five in Sub-Saharan Africa and continues to be a pressing concern for these at-risk populations [1,2]. 

II. Introduction, paragraph 2:

with potential to positively influence mothers’ health behaviors and those of their children and avert thousands of deaths annually [24]. 

III. introduction, paragraph 4:

To address preventable morbidity and mortality in mothers, infants and children through shortfalls in coverage and use of malaria interventions, we need to be able to systematically assess quality of integrated services.

2. Discussion need support from local and regional evidence.

 4.2 Public health impact

In addition to pre-existing country-specific references, we added:

paragraph 3:

Indeed, our study found that higher education was positively associated with IPTp-2 uptake in Tanzania, which is consistent with local studies that demonstrate a link between educational attainment and malaria knowledge and practice.[79,80]

Mazigo, H.D.; Obasy, E.; Mauka, W.; Manyiri, P.; Zinga, M.; Kweka, E.J.; Mnyone, L.L.; Heukelbach, J. Knowledge, Attitudes, and Practices about Malaria and Its Control in Rural Northwest Tanzania. Malar Res Treat 2010, 2010, 794261, doi:10.4061/2010/794261.

Sultana, M.; Sheikh, N.; Mahumud, R.A.; Jahir, T.; Islam, Z.; Sarker, A.R. Prevalence and associated determinants of malaria parasites among Kenyan children. Tropical Medicine and Health 2017, 45, 25, doi:10.1186/s41182-017-0066-5.

The former may reflect affluent, urban populations with improved care access and perceived lower threat of malaria as a rural disease of poverty, especially where prior interventions may have targeted rural areas [82-84].

Ameyaw, E.K.; Adde, K.S.; Dare, S.; Yaya, S. Rural-urban variation in insecticide-treated net utilization among pregnant women: evidence from 2018 Nigeria Demographic and Health Survey. Malar J 2020, 19, 407, doi:10.1186/s12936-020-03481-5.

Aberese-Ako, M.; Magnussen, P.; Ampofo, G.D.; Tagbor, H. Health system, socio-cultural, economic, environmental and individual factors influencing bed net use in the prevention of malaria in pregnancy in two Ghanaian regions. Malaria Journal 2019, 18, 363, doi:10.1186/s12936-019-2994-5.

Thank you again for your helpful review.

Round 2

Reviewer 2 Report

Thank you for making revision according to my comments in limitation.